# Unsupervised Multi-Source Federated Domain Adaptation under Domain Diversity through Group-Wise Discrepancy Minimization

## Abstract

Unsupervised multi-source domain adaptation (UMDA) aims to learn models that generalize to an unlabeled target domain by leveraging labeled data from multiple, diverse source domains. While distributed UMDA methods address privacy constraints by avoiding raw data sharing, existing approaches typically assume a small number of sources and fail to scale effectively. Increasing the number of heterogeneous domains often makes existing methods impractical, leading to high computational overhead or unstable performance. We propose GALA, a scalable and robust federated UMDA framework that introduces two key components: (1) a novel inter-group discrepancy minimization objective that efficiently approximates full pairwise domain alignment without quadratic computation; and (2) a temperature-controlled, centroid-based weighting strategy that dynamically prioritizes source domains based on alignment with the target. Together, these components enable stable and parallelizable training across large numbers of heterogeneous sources. To evaluate performance in high-diversity scenarios, we introduce Digit-18, a new benchmark comprising 18 digit datasets with varied synthetic and real-world domain shifts. Extensive experiments show that GALA consistently achieves competitive or state-of-the-art results on standard benchmarks and significantly outperforms prior methods in diverse multi-source settings where others fail to converge.

## 1 Introduction

Machine learning systems frequently face performance degradation when trained on biased or heterogeneous data sources. *Unsupervised multi-source domain adaptation (UMDA)* [1] seeks to learn models that generalize to an unlabeled target domain by leveraging labeled data from multiple sources. While multi-source setups better reflect real-world diversity, the presence of inter-source shifts introduces additional bias, making adaptation substantially more challenging. Prior UMDA research typically aligns source and target representations to improve robustness under distributional shift [2, 3, 4, 5, 6, 7, 8]. Adversarial approaches such as MCD [8] or CMSS [9] enforce consistency across domains but assume centralized data access, limiting their use in privacy-sensitive contexts such as healthcare or finance. In these settings, data protection regulations prohibit sharing raw samples, motivating *federated* and *decentralized* adaptation approaches that train collaboratively while keeping data local [10, 11, 12].

Recent federated and decentralized UMDA methods, including FADA [13], SFDA [14], KD3A [15], and FACT [16], have shown promising results on small-scale benchmarks. However, existing public benchmarks involve only a limited number of diverse source domains, leaving open the question of how these methods perform when the number of distinct and heterogeneous sources grows.

36  FACT [16], for instance, aligns random pairs of source domains in each round, computationally
37  efficient but vulnerable to high variance, while KD3A [15] performs per-domain optimization at
38  the target, achieving strong performance but with exponentially increasing cost as sources multiply.
39  These limitations underscore the difficulty current methods face in handling the growing diversity
40  and bias present in large-scale, distributed multi-source environments.

41  To address this challenge we introduce **GALA** (**G**rouping-based **A**dversarial **Lea**rning), a federated
42  UMDA framework designed to scale gracefully with high number of diverse source domains. GALA
43  integrates two key ideas: (1) an *inter-group discrepancy* objective that efficiently aligns aggregated
44  source predictions without computing all pairwise divergences; and (2) a *temperature-scaled weight-*
45  *ing* mechanism that prioritizes sources best aligned with the target. Together, these components
46  perform stable, parallelizable alignment that mitigates domain shift while preserving privacy.

47  To evaluate performance under strong domain diversity, we introduce **Digit-18**, a benchmark contain-
48  ing 18 digit datasets with diverse synthetic and real-world shifts. Across standard and large-scale
49  benchmarks, GALA consistently matches or surpasses state-of-the-art distributed adaptation methods
50  while maintaining convergence where others fail.

## 2  Methodology

51

52  **Setup.**  We consider $N$ source domains $\{\mathbb{D}_S^n\}_{n=1}^N$ with labeled samples $\{(x_i^n, y_i^n)\}$ and an unlabeled
53  target domain $\mathbb{D}_T = \{x_i^T\}$. All domains share a $C$-class task. The main objective of UMDA is to
54  learn a *feature extractor* $G : \mathcal{X} \to \mathbb{R}^d$, and a *classifier* $F : \mathbb{R}^d \to \Delta^C$, where $\Delta^C$ is the probability
55  vector over $C$ classes. Together they define the model $h = F \circ G$ that minimizes the task error
56  $\epsilon_{\mathbb{D}_T}(h) = \Pr_{(x,y) \sim \mathbb{D}_T}[h(x) \neq y]$. See [6, 17] for formal definitions of $\mathcal{H}$-divergences $d_{\mathcal{H}}, d_{\mathcal{H}\Delta\mathcal{H}}$. In
57  federated UMDA, only model updates or feature statistics can be exchanged.

58  Following classical bounds on domain adaptation [6, 17], the target error can be reduced by (1)
59  aligning feature distributions between source and target domains and (2) weighting sources according
60  to their relevance to the target. Directly computing all pairwise divergences across many domains
61  is, however, computationally infeasible. GALA addresses this by combining scalable *inter-group*
62  *alignment* with adaptive, similarity-based weighting.

### 2.1  Inter-Group Discrepancy (IGD)

63

64  Adversarial multi-source adaptation aligns source classifiers by minimizing their prediction discrep-
65  ancies on target data [16]. The naive objective

$$\mathcal{L}_{\text{full}} = \sum_{i<j} \mathbb{E}_{x \sim \mathbb{D}_T}\big[\|F_i(G(x)) - F_j(G(x))\|_1\big]$$

66  is quadratic in the number of sources $N$. FACT [16] reduces cost by sampling random domain pairs,
67  but this introduces high variance and instability as $N$ grows.

68  GALA instead partitions the sources into two disjoint groups $\mathcal{G}_1$ and $\mathcal{G}_2$, computes each group's
69  weighted average prediction, and aligns them adversarially:

$$\mathcal{L}_{\text{IGD}} = \mathbb{E}_{x \sim \mathbb{D}_T}\big[\|F_{\mathcal{G}_1}(G(x)) - F_{\mathcal{G}_2}(G(x))\|_1\big], \tag{1}$$

70  where $F_{\mathcal{G}_i} = \sum_{n \in \mathcal{G}_i} \tilde{w}_n F_n$. This group-wise formulation efficiently approximates global pairwise
71  alignment, reducing variance while remaining parallelizable. The effectiveness of IGD depends on
72  accurate source weights $\tilde{w}_n$, motivating a principled weighting mechanism.

### 2.2  Temperature-Scaled Domain Weighting (MDMGB+)

73

74  To assess source–target relevance without labels or data sharing, GALA adopts a centroid-based
75  similarity inspired by MDMGB [14]. Each domain computes class-wise feature centroids:

$$r_n^c = \frac{\sum_{x \in \mathbb{D}_S^n} \delta_c(x)\, G(x)}{\sum_{x \in \mathbb{D}_S^n} \delta_c(x)}, \qquad r_T^c = \frac{\sum_{x \in \mathbb{D}_T} \delta_c(x)\, G(x)}{\sum_{x \in \mathbb{D}_T} \delta_c(x)},$$

where $\delta_c(x)$ is the softmax output for class $c$. A cosine similarity $S(r_T^c, r_n^c)$ is computed across classes to measure alignment.

The original MDMGB averages similarities uniformly, which can overweight noisy or mismatched sources. We introduce a temperature-controlled weighting function that sharpens relevance contrast:

$$w_n = \frac{\exp(\tau\, S(r_T^c, r_n^c))}{\sum_{j=1}^N \exp(\tau\, S(r_T^c, r_j^c))}, \tag{2}$$

where $\tau > 0$ controls selectivity, larger $\tau$ emphasizes the most aligned sources. These weights are normalized within each IGD group to yield $\tilde{w}_n$. Recomputing similarities each communication round allows the weighting to adapt dynamically as representations evolve.

## 2.3 Federated Training Overview

Each round proceeds as follows: (1) the server broadcasts the global model $(G_t, F_t)$ to all sources; (2) sources compute class centroids and send them (not data) to the server; (3) the server derives similarity-based weights using Eq. 2 and aggregates feature extractors; (4) sources locally fine-tune their classifiers and return updates; (5) the server randomly partitions classifiers into two groups and updates $G$ on the target domain by minimizing Eq. 1. The process iterates until convergence. This procedure jointly reduces inter-domain bias. By combining temperature-scaled relevance estimation with group-wise alignment, GALA achieves scalable, low-variance training that preserves generalization across many heterogeneous or biased sources. A more detailed overview of the algorithm is provided in Appendix A.

## 3 Experiments

We evaluate GALA across three dimensions: (1) adaptation accuracy, (2) scalability with increasing source diversity, and (3) robustness to heterogeneous data. Comparisons include centralized baselines (MDAN [17], M$^3$SDA [9], CMSS [18], DSBN [2]) and distributed ones (SHOT [19], FADA [13], SFDA [14], FACT [16], KD3A [15]). Detailed implementation settings are provided in Appendix C.

**Datasets.** We test on (1) **Digit-Five** [9], a standard benchmark with moderate domain shift; (2) **Office-Caltech10** [20, 21], a small object benchmark; and (3) **Digit-18** (ours), a new large-scale benchmark of 18 heterogeneous digit domains created via background, color, and style augmentations to simulate strong inter-source bias. Additional details on the public benchmarks, as well as the diversity analysis, generation process, and design intuition of **Digit-18**, are provided in Appendix D.

Table 1: UMDA accuracy (%) on the Digit-Five dataset. GALA$^{\dagger}$ uses a reduced temperature $\tau = 0.2$.

| Methods | *mnist* | *mnistm* | *svhn* | *syn* | *usps* | Avg |
|---|---|---|---|---|---|---|
| Oracle | $99.5_{\pm0.08}$ | $95.4_{\pm0.15}$ | $92.3_{\pm0.14}$ | $98.7_{\pm0.04}$ | $99.2_{\pm0.09}$ | 97.0 |
| Source-only | $92.3_{\pm0.91}$ | $63.7_{\pm0.83}$ | $71.5_{\pm0.75}$ | $83.4_{\pm0.79}$ | $90.71_{\pm0.54}$ | 80.3 |
| MDAN | $97.2_{\pm0.98}$ | $75.7_{\pm0.83}$ | $82.2_{\pm0.82}$ | $85.2_{\pm0.58}$ | $93.3_{\pm0.48}$ | 86.7 |
| $M^3$SDA | $98.4_{\pm0.68}$ | $72.8_{\pm1.13}$ | $81.3_{\pm0.86}$ | $89.6_{\pm0.56}$ | $96.2_{\pm0.81}$ | 87.7 |
| CMSS | $99.0_{\pm0.08}$ | $75.3_{\pm0.57}$ | $88.4_{\pm0.54}$ | $93.7_{\pm0.21}$ | $97.7_{\pm0.13}$ | 90.8 |
| DSBN | 97.2 | 71.6 | 77.9 | 88.7 | 96.1 | 86.3 |
| FADA | $91.4_{\pm0.7}$ | $62.5_{\pm0.7}$ | $50.5_{\pm0.3}$ | $71.8_{\pm0.5}$ | $91.7_{\pm1}$ | 73.6 |
| SHOT | $98.2_{\pm0.37}$ | $80.2_{\pm0.41}$ | $84.5_{\pm0.32}$ | $91.1_{\pm0.23}$ | $97.1_{\pm0.28}$ | 90.2 |
| SFDA | 99.1 | 72.3 | 86.0 | 90.4 | 98.1 | 89.2 |
| KD3A | $99.2_{\pm0.12}$ | $87.3_{\pm0.23}$ | $85.6_{\pm0.17}$ | $89.4_{\pm0.28}$ | $\mathbf{98.5_{\pm0.25}}$ | 92.0 |
| FACT | $99.3_{\pm0.12}$ | $91.4_{\pm0.53}$ | $90.9_{\pm0.40}$ | $94.8_{\pm0.22}$ | $98.3_{\pm0.11}$ | 95.0 |
| GALA (ours) | $\mathbf{99.3_{\pm0.05}}$ | $91.0_{\pm1.34}$ | $89.7_{\pm0.02}$ | $95.0_{\pm0.08}$ | $\mathbf{98.5_{\pm0.10}}$ | 94.7 |
| GALA$^{\dagger}$ (ours) | $99.2_{\pm0.05}$ | $\mathbf{93.0_{\pm0.43}}$ | $\mathbf{91.2_{\pm0.16}}$ | $\mathbf{95.2_{\pm0.17}}$ | $98.3_{\pm0.10}$ | $\mathbf{95.4}$ |

Table 2: UMDA accuracy (%) on the Office-Caltech10.

| Methods | amazon | caltech | dslr | webcam | Avg |
|---|---|---|---|---|---|
| Oracle | 99.7 | 98.4 | 99.8 | 99.7 | 99.4 |
| Source-only | 86.1 | 87.8 | 98.3 | 99.0 | 92.8 |
| MDAN | 98.9 | 98.6 | 91.8 | 95.4 | 96.1 |
| $M^3$SDA | 94.5 | 92.2 | 99.2 | 99.5 | 96.4 |
| CMSS | 96.0 | 93.7 | 99.3 | 99.6 | 97.2 |
| DSBN | 93.2 | 91.6 | 98.9 | 99.3 | 95.8 |
| DANE | 97.4 | 97.3 | 100.0 | 100.0 | 98.7 |
| FADA | $84.2_{\pm0.5}$ | $88.7_{\pm0.5}$ | $87.1_{\pm0.6}$ | $88.1_{\pm0.4}$ | 87.1 |
| SHOT | 96.4 | 96.2 | 98.5 | 99.7 | 97.7 |
| FACT | 96.3 | 95.5 | 99.4 | 99.0 | 97.6 |
| KD3A | $\mathbf{97.4}_{\pm0.08}$ | $\mathbf{96.4}_{\pm0.11}$ | $98.4_{\pm0.08}$ | $99.7_{\pm0.02}$ | **97.9** |
| GALA (ours) | $96.5_{\pm0.19}$ | $95.0_{\pm0.17}$ | $\mathbf{100.0}_{\pm0.00}$ | $\mathbf{99.8}_{\pm0.17}$ | 97.8 |

**Standard Benchmarks.** Table 1 reports results on Digit-Five, where GALA achieves the best overall accuracy (95.4%) and outperforms all federated baselines. The temperature-scaled variant ($\tau = 0.2$, GALA[†]) further improves balance among sources. On Office-Caltech10, GALA reaches 97.8% average accuracy—comparable to KD3A (97.9%)—while achieving 100% on two domains.

**Scalability under Source Diversity.** To assess robustness as the number of sources grows, we progressively expand from 4 to 18 domains using Digit-18. Figure 1 shows that while FACT and KD3A degrade or become infeasible beyond 9 sources, GALA maintains stable accuracy through its group-wise alignment and adaptive weighting. In the full 18-domain setup (Table 3), GALA outperforms FACT by **+5.3%** on average, with the largest improvements on challenging targets such as *SVHNXS* (+9.5%) and *SYNM* (+6.4%).

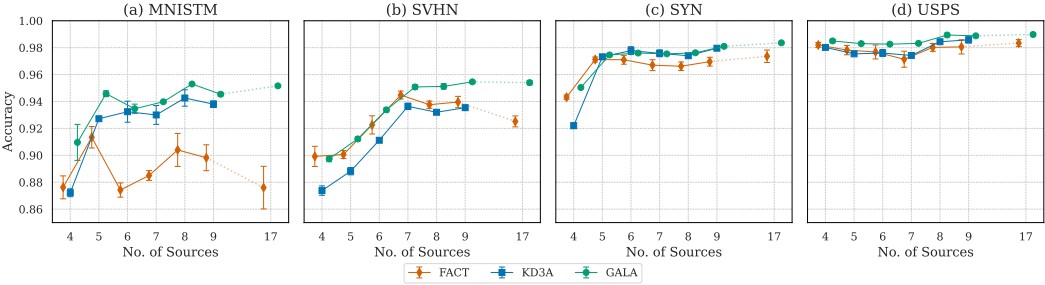

Figure 1: Performance across Digit-Five targets as the number of sources increases.

**Full Digit-18 Results.** Table 3 reports accuracy across 9 target domains in the full 18-source setting. GALA achieves a 5.3% average gain over FACT, with particularly large improvements on challenging targets such as SVHNXS (+9.5%) and SYNM (+6.4%).

Table 3: Accuracy (%) on various target domains using the Digit-18 benchmark.

| Method | mnist | mnistm | svhn | syn | usps | synm | svhn-xs | svhnstack | usps-m | Avg |
|---|---|---|---|---|---|---|---|---|---|---|
| Oracle | $99.0_{\pm0.03}$ | $95.6_{\pm0.26}$ | $88.4_{\pm0.15}$ | $97.0_{\pm0.12}$ | $98.9_{\pm0.12}$ | $83.8_{\pm0.32}$ | $84.7_{\pm0.37}$ | $86.5_{\pm0.04}$ | $92.0_{\pm0.51}$ | 91.8 |
| FACT | $98.6_{\pm0.20}$ | $87.6_{\pm1.58}$ | $92.5_{\pm0.41}$ | $97.4_{\pm0.46}$ | $98.3_{\pm0.26}$ | $79.4_{\pm1.51}$ | $79.1_{\pm4.75}$ | $91.9_{\pm1.63}$ | $87.6_{\pm1.06}$ | 87.6 |
| GALA | $\mathbf{99.3}_{\pm0.07}$ | $\mathbf{95.2}_{\pm0.10}$ | $\mathbf{95.4}_{\pm0.17}$ | $\mathbf{98.4}_{\pm0.05}$ | $\mathbf{99.0}_{\pm0.10}$ | $\mathbf{85.8}_{\pm0.23}$ | $\mathbf{88.6}_{\pm0.29}$ | $\mathbf{94.8}_{\pm0.10}$ | $\mathbf{92.9}_{\pm0.34}$ | **92.9** |

**Ablation on IGD and Weighting.** We evaluate four GALA variants on Digit-18: (i) IGD only (uniform weights), (ii) IGD + MDMGB [14], (iii) IGD + MDMGB+ ($\tau = 1.0$), and (iv) MDMGB+ without IGD. As shown in Figure 2, both components contribute positively, but their combination yields the most stable convergence and best accuracy.

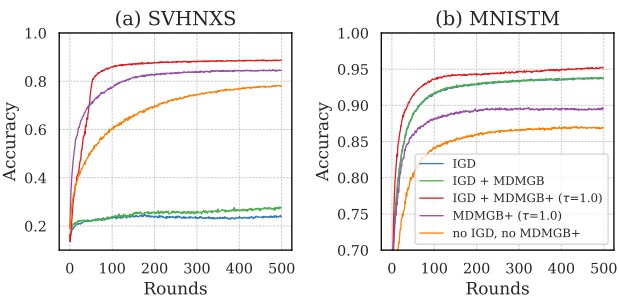

Figure 2: Ablation on IGD and weighting strategies in the 17-source Digit-18 setup.

| Source Domains | 3 | 4 | 5 | 6 | 7 | 8 | 9 |
|---|---|---|---|---|---|---|---|
| KD3A | $50.73_{\pm 0.3}$ | $103.47_{\pm 2.8}$ | $216.03_{\pm 2.7}$ | $472.22_{\pm 8.7}$ | $1029.84_{\pm 10.6}$ | $2281.38_{\pm 11.5}$ | $5600.48_{\pm 32.0}$ |
| FACT | $3.65_{\pm 0.3}$ | $3.93_{\pm 0.5}$ | $3.22_{\pm 0.4}$ | $4.06_{\pm 0.2}$ | $3.39_{\pm 0.5}$ | $3.53_{\pm 0.6}$ | $4.05_{\pm 0.5}$ |
| FedAWG | $17.27_{\pm 0.1}$ | $17.10_{\pm 0.1}$ | $17.79_{\pm 0.7}$ | $20.56_{\pm 0.5}$ | $22.21_{\pm 0.1}$ | $22.07_{\pm 0.2}$ | $22.37_{\pm 0.1}$ |

Table 4: Per-round training time (in seconds) for varying numbers of source domains.

**Runtime and Scalability.** We analyze per-round training time as the number of source domains increases (Table 4). KD3A exhibits exponential growth due to its sequential consensus stage which cannot be parallelized in practical settings, quickly becoming infeasible beyond nine sources. In contrast, both FACT and GALA maintain near-constant per-round cost through full parallelization of local updates. GALA is slightly slower per iteration than FACT due to centroid-based weighting, but its runtime scales linearly with the number of sources—avoiding the combinatorial bottleneck that limits KD3A in high-diversity settings, while still achieving comparable performance.

**Summary.** Across all benchmarks, GALA achieves state-of-the-art accuracy, stable convergence, and superior scalability. By combining inter-group alignment with temperature-scaled domain weighting, it mitigates bias from dissimilar sources and remains efficient in distributed high-diversity settings.

## 4 Conclusion

We introduced **GALA**, a federated framework for unsupervised multi-source domain adaptation that tackles the scalability challenges of diverse source settings. By combining temperature-scaled centroid-based weighting with inter-group discrepancy minimization, GALA achieves robust and efficient alignment of numerous heterogeneous source domains to an unlabeled target. It attains state-of-the-art performance on standard UMDA benchmarks and maintains strong stability and accuracy in large-scale scenarios where existing approaches degrade or fail to converge. Our new Digit-18 benchmark further validates GALA's effectiveness under realistic, high-diversity conditions.

While GALA scales well to many sources, it incurs higher per-round computational and communication costs due to full client participation. Future work could explore partial participation or reduced communication to improve efficiency. Although our experiments focus on digit datasets, Digit-18 begins to close the gap in publicly available large-scale UMDA benchmarks and provides a foundation for broader evaluation across complex visual and multimodal domains.

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

# A    GALA: Algorithm Overview

GALA combines two core components: (1) MDMGB+, which dynamically estimates domain-target similarity using centroid-based metrics and temperature-scaled softmax normalization, and (2) IGD minimization, which enables scalable adversarial alignment via randomly grouped source classifiers.

---

**Algorithm 1:** Training processes of GALA

---

**Input:** Source datasets $\{\mathbb{D}_S^n\}_{n=1}^N$, target dataset $\mathbb{D}_T$, initial model $(G, F)$, total rounds $T$, temperature $\tau$

1 **SERVER EXECUTE:**
2 **for** $t = 1, 2, \ldots, T$ **do**
3     Broadcast global model $(G_t, F_t)$ to all domains
4     **for** *class c* **in** *C* **do**
5        **foreach** *source n* **in parallel** **do**
6           $r_n^c \leftarrow \frac{\sum_{x \in \mathbb{D}_S^n} \delta_c(x) G(x)}{\sum_{x \in \mathbb{D}_S^n} \delta_c(x)}$
7        **end**
8        Target computes $r_T^c \leftarrow \frac{\sum_{x \in \mathbb{D}_T} \delta_c(x) G(x)}{\sum_{x \in \mathbb{D}_T} \delta_c(x)}$
9     **end**
10     Compute domain similarity $S(r_T, r_n)$ (Eq. **??**)
11     $w_n \leftarrow$ MDMGBPlus$(r_T, r_n)$ (Eq. 2)
12     **foreach** *source n* **in parallel** **do**
13        Initialize $(G_n, F_n) \leftarrow (G_t, F_t)$
14        Update $(G_n, F_n)$ by optimizing loss $\mathbb{E}_{(x,y) \sim \mathbb{D}_S^n}[\ell(F_n(G_n(x)), y)]$
15     **end**
16     Aggregate $G' \leftarrow \sum_n w_n G_n$ then broadcast
17     **foreach** *source n* **in parallel** **do**
18        Freeze $G'$, fine-tune $F_n$ on $\mathbb{D}_S^n$ Send updated $F_n$ to server
19     **end**
20     Randomly split sources into groups $\mathcal{G}_1, \mathcal{G}_2$
21     **foreach** *group* $\mathcal{G}_i \in \{\mathcal{G}_1, \mathcal{G}_2\}$ **do**
22        **foreach** *source* $n \in \mathcal{G}_i$ **do**
23           Compute normalized weight $\tilde{w}_n$ (Eq. **??**)
24        **end**
25        $F_{\mathcal{G}_i} \leftarrow \sum_{n \in \mathcal{G}_i} \tilde{w}_n F_n$
26        $w_{\mathcal{G}_i} \leftarrow \sum_{n \in \mathcal{G}_i} w_n$
27     **end**
28     Send $(F_{\mathcal{G}_1}, F_{\mathcal{G}_2}, G')$ to target
29     Target updates $G'$ to $G''$ by minimizing $\mathcal{L}_{\text{IGD}}$
30     Update: $G_{t+1} \leftarrow G''$;
31     Aggregate global classifier $F_{t+1} \leftarrow w_{\mathcal{G}_1} F_{\mathcal{G}_1} + w_{\mathcal{G}_2} F_{\mathcal{G}_2}$
32 **end**

---

Each round begins with the server broadcasting the global model $(G_t, F_t)$ to all domains. Domains compute class-wise centroids and upload them to the server, which calculates normalized relevance scores via MDMGB+ to weight each source's contribution.

Sources update their models locally using cross-entropy loss. The server aggregates feature extractors with similarity-based weights to form a shared extractor $G'$, sent back to sources. With $G'$ frozen, sources fine-tune classifiers and return updates to the server.

214 The server randomly partitions classifiers into two groups, averages them, and sends both groups with $G'$ to the
215 target. The target updates $G'$ to $G''$ by minimizing IGD loss between group predictions.

216 The server merges the group classifiers into a global classifier $F_{t+1}$ and sets $G_{t+1} \leftarrow G''$, updating the model as
217 $h_{t+1} = F_{t+1} \circ G_{t+1}$. This completes one round.

218 GALA tightens the generalization bound via MDMGB+ and aligns source-target representations via IGD, scaling
219 well with source diversity and consistently outperforming prior distributed UMDA methods in accuracy and
220 stability.

221 **Training Dynamics.** Figures 3 and 4 show test accuracy over training rounds. GALA converges consistently
222 across all domains, while FACT is unstable on difficult targets (MNISTM, SVHN, and SVHNXS).

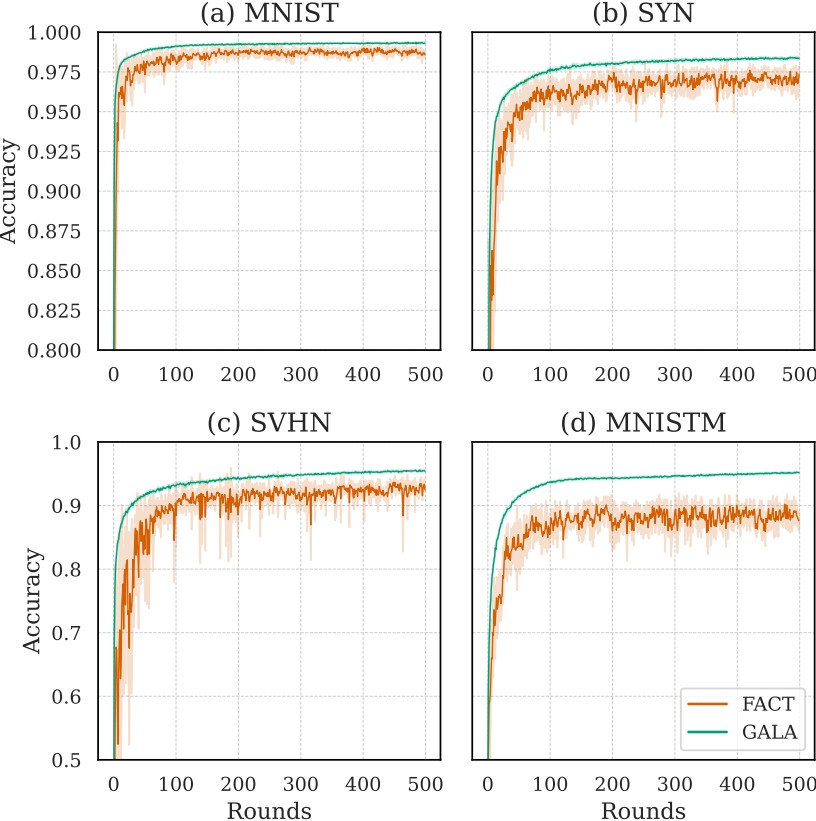

Figure 3: Test accuracy over training rounds for four Digit-Five target domains in the full Digit-18 setup.

## B   Parameter Analysis: $\tau$ in MDMGB+

224 We evaluate the sensitivity of MDMGB+ to the temperature parameter $\tau$, which controls the sharpness of source
225 relevance weighting. Figure 5 shows accuracy over training rounds for multiple $\tau$ values on two diverse Digit-18
226 targets: MNISTM and SVHNXS.

227 Very low temperatures (e.g., $\tau = 0.2$) produce overly uniform weights, limiting the model's ability to focus
228 on well-aligned sources, as seen on SVHNXS. In contrast, excessively high values (e.g., $\tau = 3$) lead to faster
229 convergence but lower final accuracy, also most evident on SVHNXS. Intermediate values ($\tau \in [0.8, 1.0]$) offer
230 the best trade-off, yielding stable and accurate performance across both target domains.

231 To further examine this effect, we repeat the analysis on the Digit-Five benchmark (Figure 6). In this smaller
232 setting with lower source diversity, we observe the opposite trend. Lower to moderate temperatures ($\tau \approx 0.4\text{--}1.0$)
233 achieve the highest accuracy, while larger values degrade performance. Overall, these results indicate that the
234 optimal $\tau$ depends on the level of domain diversity. Smaller values are preferable in low-diversity settings,
235 whereas higher values are beneficial when sources are highly heterogeneous, as in Digit-18.

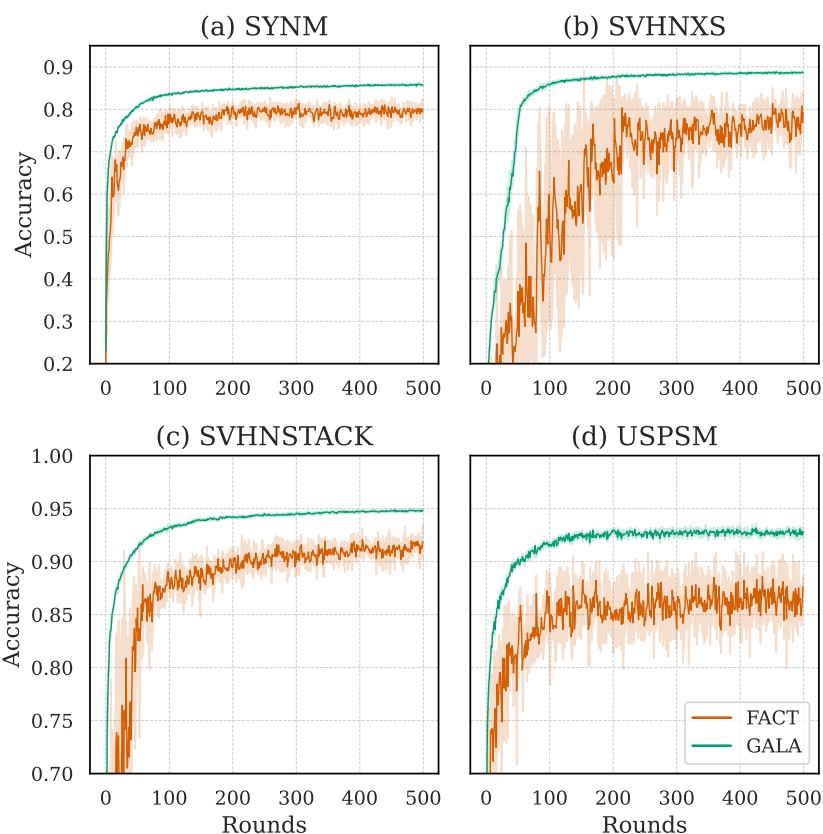

Figure 4: Test accuracy over training rounds for four Digit-18 target domains in the full Digit-18 setup.

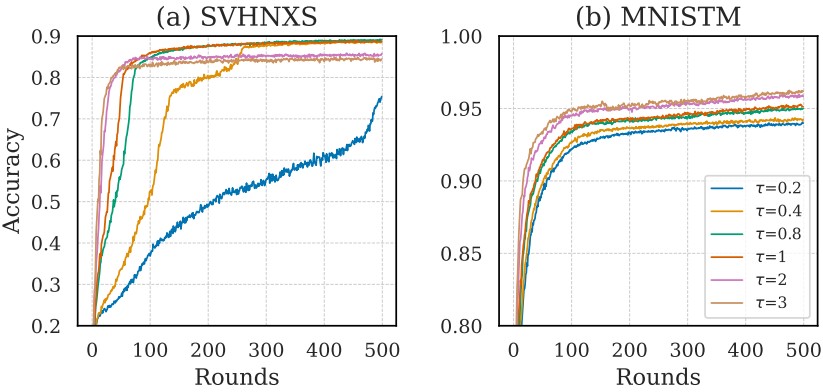

Figure 5: Effect of $\tau$ in GALA on adaptation performance for SVHNXS and MNIST-M (Digit-18).

# C   Implementation Details

We provide here the complete architectural specifications and training hyperparameters for reproducibility. The source code to reproduce all experiments will be made publicly available upon publication.

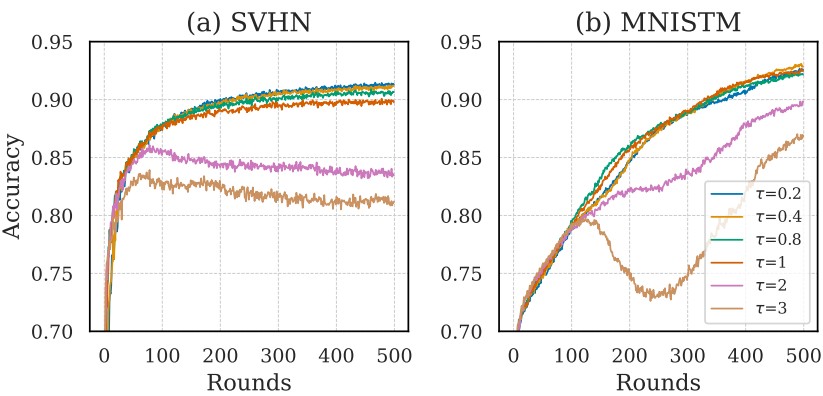

Figure 6: Effect of $\tau$ in GALA on adaptation performance for SVHN and MNIST-M (Digit-Five).

## C.1   Architectures

For experiments on Digit-Five and Digit-18, we use a lightweight 2-layer CNN as the feature extractor, followed by a 3-layer MLP classifier. The full architecture is detailed in Table 5. For Office-Caltech10, we adopt a ResNet101 backbone pretrained on ImageNet, followed by a task-specific MLP classifier as outlined in Table 6.

Table 5: Digit datasets Model Architecture

| Layer | Output Size | Kernel / Units | Details |
|---|---|---|---|
| Input | $3 \times 32 \times 32$ | - | RGB Image |
| Conv2D + BN + ReLU | $64 \times 32 \times 32$ | $5 \times 5$ | padding=2 |
| MaxPool2D | $64 \times 16 \times 16$ | $3 \times 3$ | stride=2, padding=1 |
| Conv2D + BN + ReLU | $128 \times 16 \times 16$ | $5 \times 5$ | padding=2 |
| MaxPool2D | $128 \times 8 \times 8$ | $3 \times 3$ | stride=2, padding=1 |
| Flatten | 8192 | - | |
| Dropout + FC + BN + ReLU | 3072 | - | p=0.5 |
| Dropout + FC + BN + ReLU | 100 | - | p=0.5 |
| Dropout + FC + BN + Softmax | 10 | - | p=0.5 |

Table 6: ResNet-based Predictor Architecture

| Layer | Output Size | Units | Details |
|---|---|---|---|
| ResNet101 Backbone | 1000 | - | Pretrained on ImageNet |
| Dropout + FC + BN + ReLU | 500 | - | p=0.5 |
| FC + BN + Softmax | {10, Number of Classes} | - | Task-specific classes |

## C.2   Training Details

Table 7 summarizes the training parameters used for each benchmark. All models are trained using SGD with momentum 0.9 and weight decay $5 \times 10^{-4}$. We set the batch size to 128 and train for 500 rounds. Communication occurs once per round ($r = 1$), and each training phase (source training, fine-tuning, adversarial alignment) is performed for one epoch. Following [15], we apply mixup augmentation ($\alpha = 0.2$) for Office-Caltech10 only.

## C.3   Hardware

All experiments were run on a compute node with an AMD EPYC 7713 64-core CPU and a single NVIDIA A100 GPU (40GB).

Table 7: Implementation details of our GALA on three benchmark datasets.

| Parameters | Digit-Five | Digit-18 | Office-Caltech10 |
|---|---|---|---|
| Data Augmentation | None | | Mixup ($\alpha = 0.2$) |
| Backbone | 2-layer CNN | | ResNet101 (pretrained=True) |
| Optimizer | SGD with momentum = 0.9 and weight decay =$5 \times 10^{-4}$ | | |
| Learning Rate Schedule | CustomLR ($\gamma$=0.75) | | ExponentialLR ($\gamma$=0.9) |
| Batch Size | 128 | | |
| Total Rounds | 500 | | |
| Communication Rounds | $r = 1$ | | |
| Temperature | $\tau = 0.2$ | | $\tau = 1.0$ |

# D Datasets

**Office-Caltech10.**    Office-Caltech10 consists of for domains: Amazon, Webcam, DSLR and Caltech. The images show objects from 10 different classes which are shared between Office [20] and Caltech-265 [21] datasets.

**Digit-Five.**    The Digit-Five dataset [3] is a popular benchmark for digit recognition. It consists of the following five datasets, each representing a separate domain: MNIST, MNIST-M, Street-View House Numbers (SVHN), Synthetic Digits (SYN), and USPS.

## D.1 Digit-18 Benchmark

**Digit-18** is our proposed large-scale benchmark composed of 18 domains, created by applying systematic transformations to existing digit datasets. It is specifically designed to evaluate the robustness and scalability of UMDA methods in diverse high-source scenarios. Our goal was to ensure sufficient variability and domain shifts across the domains. Thus, we did not apply each transformation to every dataset, as some domains are already similar. Sample images from each domain are shown in Figure 7.

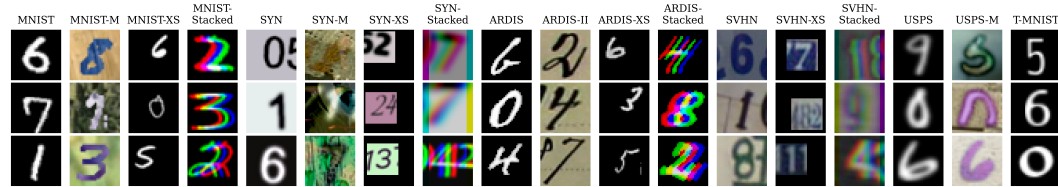

Figure 7: Sample images from each domain in the **Digit-18** benchmark.

### D.1.1 Base Datasets

- **ARDIS** [22]: A historical handwritten digit dataset extracted from Swedish church records. We use 6,600 training and 1,000 testing samples. We include two variants: a normalized version (matched to MNIST) and an unprocessed version with original grayscale backgrounds and image noise, referred to as **ARDIS II**.

- **TMNIST** [23]: Typography-MNIST contains 22,400 training and 7,500 test images of digits rendered in various fonts. The images are grayscale on a black background, similar to MNIST but with greater stylistic diversity.

### D.1.2 Domain Transformations

We applied the following transformation strategies to simulate diverse and challenging domain shifts:

- **Background Augmentation:** Following MNIST-M [24], we overlay complex colored backgrounds on digit images from SYN, SVHN, and USPS to create SYNM, SVHNM, and USPSM.

- **Scaling:** Original digit images are resized to $20 \times 20$ and re-centered on a $32 \times 32$ black canvas. Applied to MNIST, SYN, SVHN, and ARDIS, yielding *-XS domains (e.g., MNISTXS).

- **Stacking:** We introduce pixel-level channel misalignments by shifting R, G, B channels in opposite directions. Applied to grayscale domains, this generates color interference effects. Used for MNIST, SYN, SVHN, and ARDIS to generate *-STACK domains.

### D.1.3 Domain Shift Analysis

To assess domain similarity and difficulty, we trained simple models on each domain independently and evaluated them across all other domains. These models used the same architecture and training settings as in the UFDA experiments (500 epochs, SGD with momentum 0.9, fixed learning rate 0.001). The accuracy matrix in Figure 8 reveals cross-domain generalization trends and helps characterize inter-domain shifts.

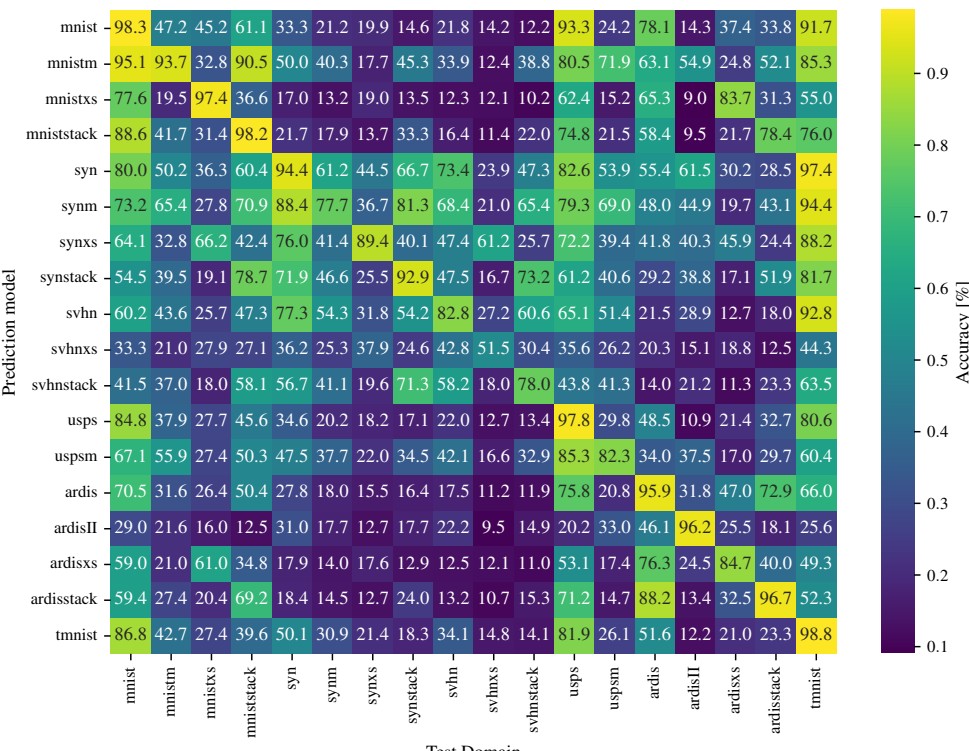

Figure 8: Cross-domain similarity matrix: each row corresponds to a model trained on a source domain and evaluated on all target domains.

Notably, models trained on clean datasets (e.g., MNIST) fail to generalize well to complex variants (e.g., SYNM), while models trained on background-augmented domains (e.g., MNISTM) transfer better to simpler settings. These insights informed the domain selection process and help contextualize results in our experiments.

## E  Robustness Analysis of FACT

FACT randomly selects two source domains in each communication round to perform inter-domain distance minimization. This random pairing strategy introduces instability in training, as the model update becomes highly sensitive to the selected source combination. We observed frequent fluctuations in test accuracy, especially on more challenging target domains.

To investigate this further, we analyze training behavior on the Digit-18 benchmark. Figure 9 shows round-to-round changes in test accuracy for two particularly difficult target domains: SVHNXS and MNISTM. We annotate the selected source domain pairs in the rounds with the largest single-round accuracy increases and decreases. To focus on model behavior during convergence, we restrict this analysis to the phase after the first 110 communication rounds (i.e., after warm-up).

For SVHNXS, the largest accuracy drops occur when both selected sources are *-M domains, all of which score below 25 % similarity with SVHNXS. Because SVHNXS consists of black-and-white digit images with extensive black backgrounds, it cannot leverage the colorful backgrounds of *-M sources, resulting in negative transfer. Notably MNISTM paired with SYNXS lead to marked performance improvements, indicating that a dissimilar source can still be beneficial when paired with a complementary one. Various *-XS sources, especially SYNXS produce significant accuracy gains. This improvement can be explained by the similar data generation processes of these domains, which help the model capture SVHNXS's characteristics. SYNXS, the most similar

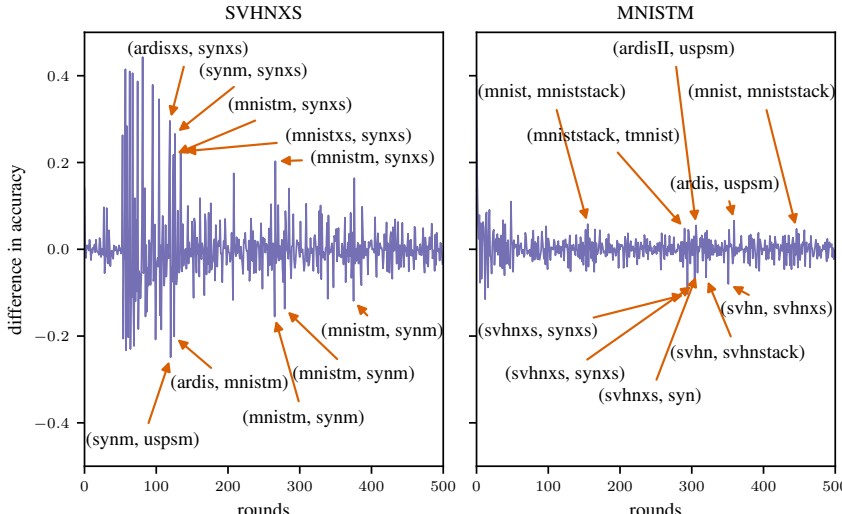

Figure 9: Round-to-round accuracy difference of FACT. Clients associated with the highest single-round accuracy increases and decreases are annotated.

source to SVHNXS, appears in all beneficial pairs. Note that it is also consistently most highly weighted by
GALA, demonstrating our method's ability to identify the most relevant sources. Similarly, for MNISTM,
selecting MNIST-like and -M domains leads to test accuracy improvements compared to the previous round. In
contrast, selection of SVHN-* domains and SYNXS, with similarity scores below 50%, results in significant
accuracy drops.

These observations highlight the robustness issues of FACT's random source selection mechanism in large
multi-source settings.

## F  Adaptive Source Weighting in GALA

GALA dynamically assigns a weight to each source client in every training round, determining its influence
on the shared model. Figure 10 illustrates the evolution of these weights for the target domains MNISTM and
USPS in the Digit-18 setting. The top five most highly weighted sources are highlighted.

For MNISTM, the top-weighted domains align with those found most similar in our similarity analysis. These
include MNIST-like and *-M datasets, supporting the expectation that their combination is well-suited for
learning MNISTM. The USPS plot illustrates the value of dynamic re-weighting. In early rounds, simpler
domains like MNIST and TMNIST dominate. Later, the weights shift toward USPSM and SYN, reflecting a
focus on learning finer details and USPS-specific features.

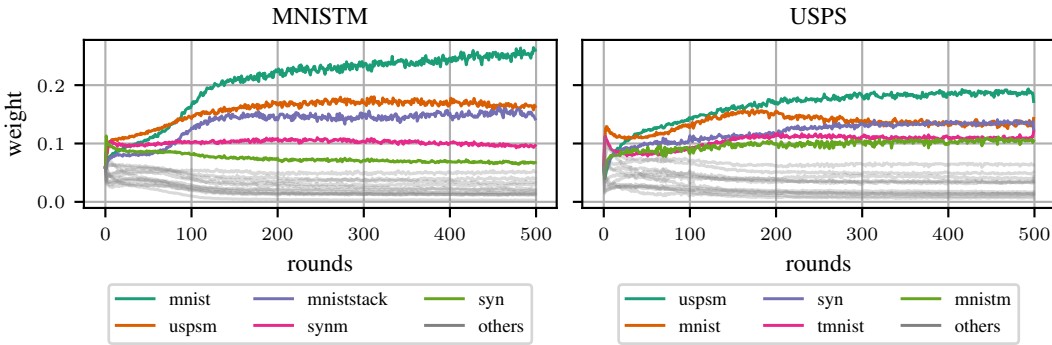

Figure 10: Evolution of source weights assigned by MDMGB+ for MNISTM and USPS (Digit-18
setting). The five most highly weighted source domains are highlighted. Shifts over time reflect both
domain similarity and the model's adaptation to target-specific learning needs.