# OpenReview forum: "Unsupervised Multi-Source Federated Domain Adaptation under Domain Diversity through Group-Wise Discrepancy Minimization"
_EurIPS.cc/2025/Workshop/UPLB — UPLB2025_

### Official Review · Reviewer_oNFn · 2025-10-27
**Review Multi-source federated domain**

**Rating:** 6
**Confidence:** 2

**Review:**

This work deals with Machine Learning tasks based on heterogeneous data-sources to perform domain adaptation. The main point of the method is to propose an approach that can deal with numerous different sources, at difference with previous methods. In addition, they propose a new benchmark for based on handwritten digits using 18 heterogenous digit domains.

The work proposes a new objective function for domain adaptation, based on weighted representation of the different sources, and use a centroid-based similarity to compute the weights of each source.

The method has in general very good performance in the benchmark. My only concern is the alignment with the workshop's thematic. The proposed method can be understood as part of tackling the "dataset shift" scenario, and multi-source tasks can enter into handling a variety of sources.

---

### Decision · Program_Chairs · 2025-11-03

Accept (Poster)